# Molecular Characterization and Pathogenicity of *Alternaria* spp. Associated with Black Rot of Sweet Cherries in Italy

**DOI:** 10.3390/jof9100992

**Published:** 2023-10-07

**Authors:** Muhammad Waqas, Simona Prencipe, Vladimiro Guarnaccia, Davide Spadaro

**Affiliations:** 1Department of Agricultural, Forest and Food Sciences (DISAFA), University of Torino, Largo Braccini 2, 10095 Grugliasco, TO, Italy; muhammad.waqas@unito.it (M.W.); simona.prencipe@unito.it (S.P.); vladimiro.guarnaccia@unito.it (V.G.); 2AGROINNOVA—Interdepartmental Centre for Innovation in the Agro-environmental Sector, University of Torino, Largo Braccini 2, 10095 Grugliasco, TO, Italy

**Keywords:** *Alternaria alternata*, *Alternaria arborescens* species complex, *Prunus avium*, phylogeny

## Abstract

Black rot is limiting the production of sweet cherries in Italy. Dark brown to black patches and sunken lesions on fruits are the most common symptoms of *Alternaria* black rot on sweet cherry fruits. We isolated 180 *Alternaria* spp. from symptomatic cherry fruits ‘Kordia’, ‘Ferrovia’, and ‘Regina’ harvested in Northern Italy, over three years, from 2020 to 2022. The aim was to identify and characterize a selection of forty isolates of *Alternaria* spp. based on morphology, pathogenicity, and combined analysis of rpb2, Alt-a1, endoPG and OPA10-2. The colonies were dark greyish in the center with white margins. Ellipsoidal or ovoid shaped conidia ranging from 19.8 to 21.7 μm in length were observed under a microscope. Based on the concatenated session of four gene regions, thirty-three out of forty isolates were identified as *A. arborescens* species complex (AASC), and seven as *A. alternata*. Pathogenicity was evaluated on healthy ‘Regina’ sweet cherry fruits. All the tested strains were pathogenic on their host. This study represents the first characterization of *Alternaria* spp. associated with black rot of cherries in Italy and, to the best of our knowledge, it is also the first report of AASC as an agent of black rot of sweet cherries in Italy.

## 1. Introduction

Sweet cherry (*Prunus avium* L.) is an economically important stone fruit which belongs to the genus *Prunus* within the Rosaceae family. Italy is the seventh sweet cherry producer in the world after Turkey, United States of America, Chile, Iran, Uzbekistan, and Spain and the second in Europe [1]. In Italy, sweet cherry is cultivated on an area of 28,609 ha, with an annual production of approximately 107,905 t [2].

Sweet cherries are highly perishable fruit, with a shelf life of 7–14 days [3] and they are susceptible, during cold storage, to decay caused by postharvest pathogens, including *Monilinia* spp., agent of brown rot; *Botrytis cinerea,* agent of gray mold; *Penicillium expansum*, agent of blue mold; and *Rhizopus* spp., agent of soft rot [4,5,6,7]. Cherry production could be threatened also by black rot caused by *Alternaria* species, which is emerging as a major fungal disease [8,9,10]. Symptoms of black rot of *Alternaria* appear as dark brown to black patches on the outer surface of fruits. These patches gradually increase in size and surface and, under high-humidity conditions, may develop white to light brown, fluffy and moldy growth, and ultimately can cause complete fruit rot [8,11,12].

The genus *Alternaria* was originally described by Nees von Esenbeck in 1816; it is part of the *Pleosporaceae* family [13] and includes endophytic, pathogenic, and saprobic species [14,15,16,17]. *Alternaria* spp. are distributed all over the world and may infect over 4000 plant species [13,15,17], such as vegetables, cereals and fruit trees in field and during storage [18,19,20,21,22,23]. Black rot caused by *A. alternata* (Fr.) Keissl. was recently reported on cherries in Chile [9] and in China [8,10]. *A. alternata* is also associated with brown to black spots on cherry leaves [24,25]. Alternaria rots and spots are becoming more and more frequent on cherry and on other fruit crops due to climate change, characterized by the increase of average temperature, high-humidity conditions and stressed plants, weakened by abiotic stresses. Biological and environmental factors are causing a shift in the fruit microbiological ecology, which is influencing the pre- and postharvest development of *Alternaria* spp. [26].

In the past, *Alternaria* spp. were identified based on macro- and micro-morphological characteristics [16,27]. Nowadays, DNA-based molecular techniques are used to identify *Alternaria* at species level to avoid morphological variations depending on the environmental conditions [13,15,28,29,30,31,32]. Different molecular approaches have been used for the identification of *Alternaria* section, resulting in several taxonomic revisions [13,16,28,33,34,35,36,37]. Most small-spored *Alternaria* species with concatenated conidia belong to the section Alternaria [16], and *A. alternata* (Fr.) Keissl. and *A. arborescens* Simmons are important representative plant pathogens of this section [16]. To the best of our knowledge, *Alternaria* spp. were not previously reported and characterized as agents of black rot of sweet cherries in Italy.

The aims of this study are to isolate and identify the *Alternaria* species associated with black rot of sweet cherries, based on morphological and phylogenetic analysis, and to confirm their pathogenicity and virulence on sweet cherries.

## 2. Materials and Methods

### 2.1. Sampling and Isolation

A monitoring of postharvest diseases was conducted in sweet cherry harvested from orchards located in Piedmont, Northern Italy, from 2020 to 2022. A total of 180 isolates were isolated from symptomatic cherry fruits belonging to the varieties ‘Regina’, ‘Kordia’, and ‘Ferrovia’ in the packinghouses of Piedmont (Figure 1A,B). Sweet cherry fruits were surface disinfected with 1% sodium hypochlorite for 1 min, rinsed in sterile water for 1 min, and dried on sterile filter paper. Then five pieces of black rotten fruits were cut at the margin between healthy and infected tissues and plated on potato dextrose agar (PDA, VWR international, Leuven, Belgium) [29] containing streptomycin (0.025 g/L). The PDA plates were incubated at 22 ± 1 °C for 3 days. Pure cultures were obtained by transferring the mycelium plug from the edge of the colonies and placed in fresh PDA plates. Isolates used in this study were maintained and kept at −80 °C in the culture collection of the University of Turin, Torino, Italy (Table 1).

### 2.2. Micro and Macro-Morphological Characteristics

*Alternaria* isolates were plated on PDA medium (in triplicate) and incubated at 25 ± 1 °C for the macro-morphological characteristics (color, margin, diameter, and texture), according to Simmons et al. [27]. Mean radial growth was measured after 7 days of incubation (Table 1). For the microscopic features (conidia and conidiophore) the isolates were grown onto Potato Carrot Agar (PCA, HiMedia Laboratories, Mumbai, India) under 12 h light and 12 h dark cycle for 15 days [27]. Thirty conidia per isolate were examined using an Eclipse 55i microscope (Nikon, Tokyo, Japan) at 40× magnification.

### 2.3. DNA Extraction and PCR Amplification

Genomic DNA of 40 *Alternaria* isolates was extracted with an E.Z.N.A. Fungal DNA mini kit (Omega Bio-tek, Darmstadt, Germany) from 0.1 g mycelium of 7-day-old culture grown on PDA (VWR international) according to the manufacturer’s instructions. The quality and concentration of extracted DNA was determined using NanoDrop 2000 spectrophotometer (Thermo scientific, Wilmington, DE, USA). The primers Alt-for and Alt-rev [38] were used to amplify part of Alternaria major allergen gene (Alt-a1). The partial *endopolygalacturonase* gene (endoPG) was amplified using PG3 and PG2b primers [39,40]. The primer sets RPB2-5f2 and fRPB2-7cr [41,42] were used to amplify the part of RNA polymerase second largest subunit (rpb2). The primers OPA 10-2R and OPA 10-2L [28] were used to amplify part of an anonymous gene region (OPA10-2). The amplification of all four loci were performed according to PCR amplified conditions described by Prencipe et al. [43]. The PCR cycling conditions adopted for rpb2 and Alt-a1 were described by Woudenberg et al. [44], and for endoPG and OPA 10-2, by Andrew et al. [28]. The amplification products were analyzed on 1% agarose (VWR International, Milan, Italy) after staining with GelRedTM. PCR products were purified with the PCR Purification Kit (QIAquick^®^, Hilden, Germany) following manufacturer instructions, before sequencing by Macrogen Europe B. V. (Amsterdam, The Netherlands).

### 2.4. Phylogenetic Analysis

Phylogenetic analysis was performed using sequences generated in this study and reference sequences of *Alternaria* spp. [32] (Appendix A). After cutting the trimmed regions in Geneious v. 11.1.5 program (Auckland, New Zealand) and manual correction in MEGA v. 7, a dataset 2229 bp of 281 bp for Alt-a1, 479 bp for endoPG, 634 bp for OPA10-2, and 835 bp for rpb2 was obtained. The sequences were aligned using CLUSTALW in MEGA v. 7 [45]. Phylogenetic analysis was performed using the concatenated dataset (rpb2, OPA 10-2, Alt-a1 and endoPG) for the identification of *Alternaria* isolates at species level. The phylogeny was based on maximum parsimony (MP) and Bayesian inference (BI) used in a concatenated analysis. For BI, the best fit evolutionary model for each partitioned locus was estimated using MrModeltest v. 2.3 [46] and incorporated into the analysis. MrBayes v. 3.2.5 [47] was used to generate phylogenetic trees under optimal criteria per partition. The Markov chain Monte Carlo (MCMC) analysis used four chains and started from a random tree topology. The heating parameter was set at 0.2 and trees were sampled every 1000 generations. The analysis stopped when the average standard deviation of split frequencies was below 0.01. The MP analysis was performed using Phylogenetic Analysis using Parsimony (PAUP) v. 4.0b10 [48]. Phylogenetic relationships were estimated by heuristic searches with 100 random addition sequences. Tree bisection-reconnection was used, with the branch swapping option set on ‘best trees’, with all characters equally weighted and alignment gaps treated as fifth state. Tree length (TL), consistency index (CI), retention index (RI) and rescaled consistence index (RC) were calculated for parsimony, and the bootstrap analysis [49] was based on 1000 replications. Sequences generated in this study were deposited in GenBank (Appendix A).

### 2.5. Pathogenicity Assay

The same isolates used for molecular analysis were used for the pathogenicity test. Conidia of tested *Alternaria* isolates, produced on 21-day-old PDA cultures incubated at 22 ± 1 °C under a 12 h photoperiod, were used to obtain conidial suspensions, which were prepared by scrapping off the conidia from the surface of PDA plates with sterile water and 10 μL of Tween 20, which were then filtered through four layers of sterile gauze, as described by Prencipe et al. [43]. The concentration of the collected spore suspension was adjusted to 10^5^ conidia/mL. Pathogenicity tests were conducted on healthy fruits of cherries ‘Regina’ using the method described by Ahmad et al. [8]. Fruit surfaces were disinfected with 1% sodium hypochlorite for 1 min, washed with distilled water and air dried for 5 min. The experiment was conducted using ten fruits per isolate. Each fruit was inoculated with 1 µL of conidial suspension by creating one wound (1 mm^2^) with a sterile needle. Control fruits were treated with sterilized distilled water. Inoculated fruits were placed in plastic trays and covered with a plastic film and incubated at 20 ± 2 °C until symptoms appeared. After 10 days of inoculation, rot diameters were measured. Re-isolations were conducted from the inoculated fruits as described above. Each isolate was tested twice.

### 2.6. Statistical Analysis

The statistical analysis was performed using SPSS software (IBM SPSS Statistics v. 28.0.1.0). Rot diameters obtained in the pathogenicity test were subjected to the analysis of variance (ANOVA) and the mean values were separated by Tukey test (*p* ≤ 0.05).

## 3. Results

### 3.1. Fungal Isolation, Identification, and Morphological Characterization

Typical symptoms of dark brown to black patches and sunken lesions were observed on sweet cherry fruits. Out of 180 isolates obtained from symptomatic sweet cherry fruits, 40 isolates were identified as *Alternaria* spp. based on micro and macro-morphological observations [27] (Table 1).

Isolates of *Alternaria* spp. isolated from infected sweet cherry fruits were identified based on their morphological characteristics [27]. All the *Alternaria* isolates formed aerial mycelium on PDA, and most of the colonies were dark greyish in the center with white margins after 7 days of incubation at 25 ± 1 °C under 12 h light and dark cycle. The mycelium becomes dark brown after 10 to 14 days. The mean radial growth of colony was 4.96 ± 0.52 cm (Table 1, Figure 1C,D). The conidiophores were light brown and the conidia were ellipsoidal or ovoid with 1–4 transverse septa, a mean of 21.01 ± 0.39 μm in length and 11.11 ± 0.26 μm in width (Table 1, Figure 1E,F).

### 3.2. Phylogenetic Analysis

The sequences obtained in this study were subjected to a BLAST search in NCBI’s GenBank (https://blast.ncbi.nlm.nih.gov/Blast.cgi; accessed on 10 January 2023) nucleotide database for preliminary identification. A multilocus phylogenetic analysis was conducted using forty isolates of *Alternaria* spp. based on the sequences from four genes (rpb2, OPA 10-2, Alt-a1 and endoPG) and sixty-two reference sequences [32], including the outgroup *Alternaria nobilis*. A total of 184 nucleotides were parsimony-informative, 276 were variable and parsimony-uninformative, and 1769 were constant. A maximum of 1000 equally maximum parsimony (MP) trees were saved (Tree length = 771, CI = 0.642, RI = 0.864 and RC = 0.555). Bootstrap support values from the MP analysis are incorporated on the Bayesian tree in Figure 2. For the Bayesian analyses (BI), MrModeltest suggested that all partitions should be analyzed with Dirichlet state frequency distributions. The following models were recommended by MrModeltest and used: K80 for Alt-a1, SYM + G for endoPG and *rpb2*, and K80 + I + G for OPA 10-2. In the BI, the Alt-a1 partition had 80 unique site patterns, the endoPG partition had 92 unique site patterns, the OPA 10-2 partition had 136 unique site patterns, the *rpb2* partition had 155 unique site patterns and the analysis ran for 6,595,000 generations, resulting in 6596 trees, of which 4947 trees were samples used to calculate the posterior probabilities. Based on multilocus phylogenetic analysis, thirty-three out of forty strains clustered with AASC, whereas the remaining seven strains belonged to *A. alternata* (Figure 2 and Table 1).

### 3.3. Pathogenicity Assay

The tested *Alternaria* strains were pathogenic on artificially inoculated sweet cherries. Symptoms appeared as the development of black rotted spots and sunken lesions all over the inoculation point after 7 days post inoculation (Figure 3). Lesions increased in diameter with disease progress and morphologically appeared similar to those observed in naturally infected fruits (Figure 3). In our pathogenicity, no significant differences were observed among the *Alternaria* strain/species tested combinations. Only two strains (Ch4 and Ch17) from AASC showed significant differences: the isolate Ch-17 showed the highest rot diameter (17.0 ± 2.13 mm), whereas the isolate Ch-4 showed the lowest rot diameter (8.43 ± 2.03 mm) (Table 1). To fulfill Koch’s postulates, re-isolation was carried out on all the symptomatic inoculated fruits, and isolates were identified as *Alternaria* spp. using the rpb2 gene [41]. Healthy control fruits did not develop any symptoms.

## 4. Discussion

In our study, isolates of *Alternaria* spp. associated with black rot of sweet cherries in Italy were identified as *A. alternata* and *A. arborescens* species complex (AASC) based on morphological and phylogenetic analysis at species level. During three years of investigation, all orchards showed the presence of *A. alternata* and AASC. In the present study, AASC was the predominant fungus isolated from rotted cherries. Along with *Alternaria* spp., we found the co-occurrence of other secondary fungal pathogens associated with rotten fruits, including *Cladosporium* spp., *Monilinia* spp., *Penicillium* spp. and *Botrytis cinerea*. In several studies, AASC has been reported as a causal agent of black rot of mandarin [50], blueberry [51], pomegranate fruit [52], and Japanese plum [53]; core rot of apple [54], leaf blotch and fruit spot diseases of apple [55]; black spots on fruit and leaves of pear [40]; and brown spots on sweet orange and lemon in Italy [56]. No information is reported about the association of AASC with sweet cherries. In contrast, *A. alternata* has been reported as being associated with postharvest rot of sweet cherry [15], black spot of cherry fruits in China [10], and leaf spot disease of sweet cherry in Greece [25] and Turkey [24]. Furthermore, *A. alternata* has been reported as the main causal agent of black rot of cherry tomatoes [57], and fruit rot on strawberry [51,58], mandarin [50], pomegranate fruit [52,59], Japanese plum [53], *Solanum muricatum* [60], and opium poppy [61]. Most studies did not report *Alternaria* spp. as a main pathogen, but in our study, we isolated *Alternaria* spp. as a major pathogen from sweet cherry fruits. In our study, all isolates of *Alternaria* spp. were identified based on cultural and morphological characteristics, as they all formed conidia, similar to the observations of Prencipe et al. [43] and Şimşek et al. [24]. Our *Alternaria* isolates were dark grayish with white margins and formed aerial mycelium on PDA, with an average size of conidia (19.7–21.7 × 10.8–11.6) on PCA. These morphological and cultural characteristics were similar to previous studies [40,62], but different from those described by Şimşek et al. [24]. This phenomenon could be due to the morphological plasticity of *Alternaria* spp. [17] and the fact that the morphology of conidia is dependent on conidial age and culture conditions [37].

The taxonomy of small-spored *Alternaria* spp. suffered from controversies because *Alternaria* spp. shared similar morphological characteristics and the size of conidia [33,50,63]. Molecular-based assays could be used for the correct identification of *Alternaria* spp. along with morphological characteristics [32,33,64]. Molecular analysis also had some challenges to overcome, as *Alternaria* section *Alternaria* cannot be recognized using standard genetic loci due to the little or no variation in molecular markers [13,28,36,43]. Previous studies suggested that the identification criteria for low resolution of species delimitation in small-spored *Alternaria* spp. are only significant when employing the combination of different genes together [14,17,27]. To overcome the issues, the phylogeny of *Alternaria* sections was solved by using nine gene regions (Alt a 1, endoPG, gapdh, ITS, LSU, OPA10-2, rpb2, SSU and tef1) by Woudenberg et al. [32]. The concatenated session of six gene regions of Alt a 1, endoPG, ITS, OPA10-2, rpb2, and tef1 was able to separate AASC from *A. alternata* [14]. In our study, we excluded the slowly evolving gene tef 1 and used the genes proposed by Prencipe et al. [43]. Moreover, previous studies confirmed that alt a1 [65] and OPA10-2 [43] were sufficient to separate the *A. alternata* from AASC. The four loci (Alt-a1, endoPG, opa10-2 and rpb2) used for the phylogenetic analysis performed in this research allowed us to identify thirty-three isolates as members of AASC, and seven isolates as *A. alternata*. The combined analysis of phylogenetic trees showed similarity with previous studies [14,43]. However, the phylogeny of our study obtained from the concatenated session of four genes has low bootstrap value in agreement with previous studies [35,53]. Considering our phylogenetic analysis results, the inclusion of more genes, such as gaphd, LSU and SSU, could increase the discrimination power, as previously proposed by Zhang et al. [17].

In our pathogenicity test, sweet cherry fruits were wounded and inoculated with conidial suspension of *Alternaria* strains. Pathogenicity results showed that all the *Alternaria* strains were pathogenic, had high virulence, and produced irregular lesions when fruits were inoculated by conidial suspension. According to previous studies, *A. alternata* species were pathogenic on sweet cherry fruits inoculated with conidial suspension and lesions were observed [8]. Additionally, Prencipe et al. [43] reported that isolates of *A. alternata* and AASC were pathogenic when inoculated with conidial suspension on wounded European pear. In previous studies, *Alternaria* spp. was confirmed to be an opportunistic, saprophytic and weak pathogen that enters the plant tissues through natural openings and wounds [24,66,67], when the plant becomes more susceptible to diseases [68]. AASC strains caused lesions ranging from 8.43 to 17.0 mm in size while the lesion size of *A. alternata* strains ranged from 9.06 to 13.81 mm. According to previous studies [55,69], pathogenicity may be isolate-dependent instead of species-dependent. Our results showed that there was little difference among the tested *Alternaria* spp.

In conclusion, the present work describes for the first time the presence of AASC as an agent of black rot on sweet cherry fruits in Italy. Investigation should verify if other species of *Alternaria* could be involved in black rot of cherry fruit. Moreover, more isolates from different geographical areas should be included to explore the genetic diversity of the causal agents of black rot of cherry. Future studies will focus on developing and testing effective disease management strategies both in field and during postharvest. Furthermore, future studies will focus on characterizing the mycotoxin production potential of *Alternaria* species on sweet cherries.

## Figures and Tables

**Figure 1 jof-09-00992-f001:**
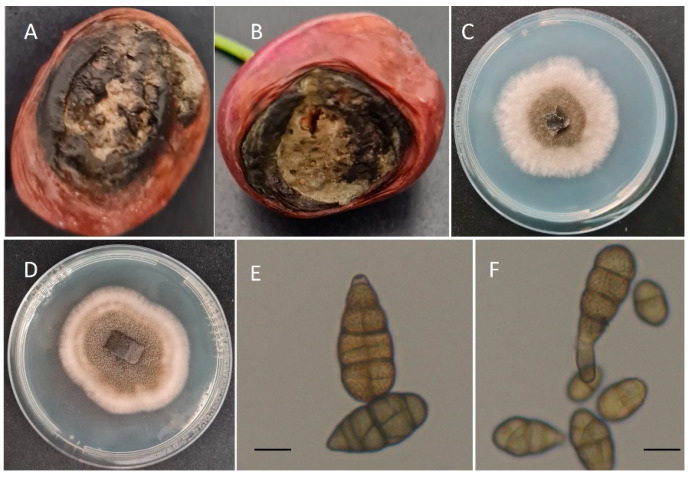
(**A**,**B**) Symptoms of *Alternaria* black rot on naturally infected sweet cherry cv. Regina; (**C**) colony growth of AASC (strain T8) after 7 days on PDA; (**D**) colony growth of *A. alternata* (strain GR13) after 7 days on PDA; (**E**) conidia of AASC (strain T8) obtained after 15 days on PCA; (**F**) conidia of *A. alternata* (strain GR13) obtained after 15 days on PCA. Scale bar: (**E**,**F**) = 10 µm.

**Figure 2 jof-09-00992-f002:**
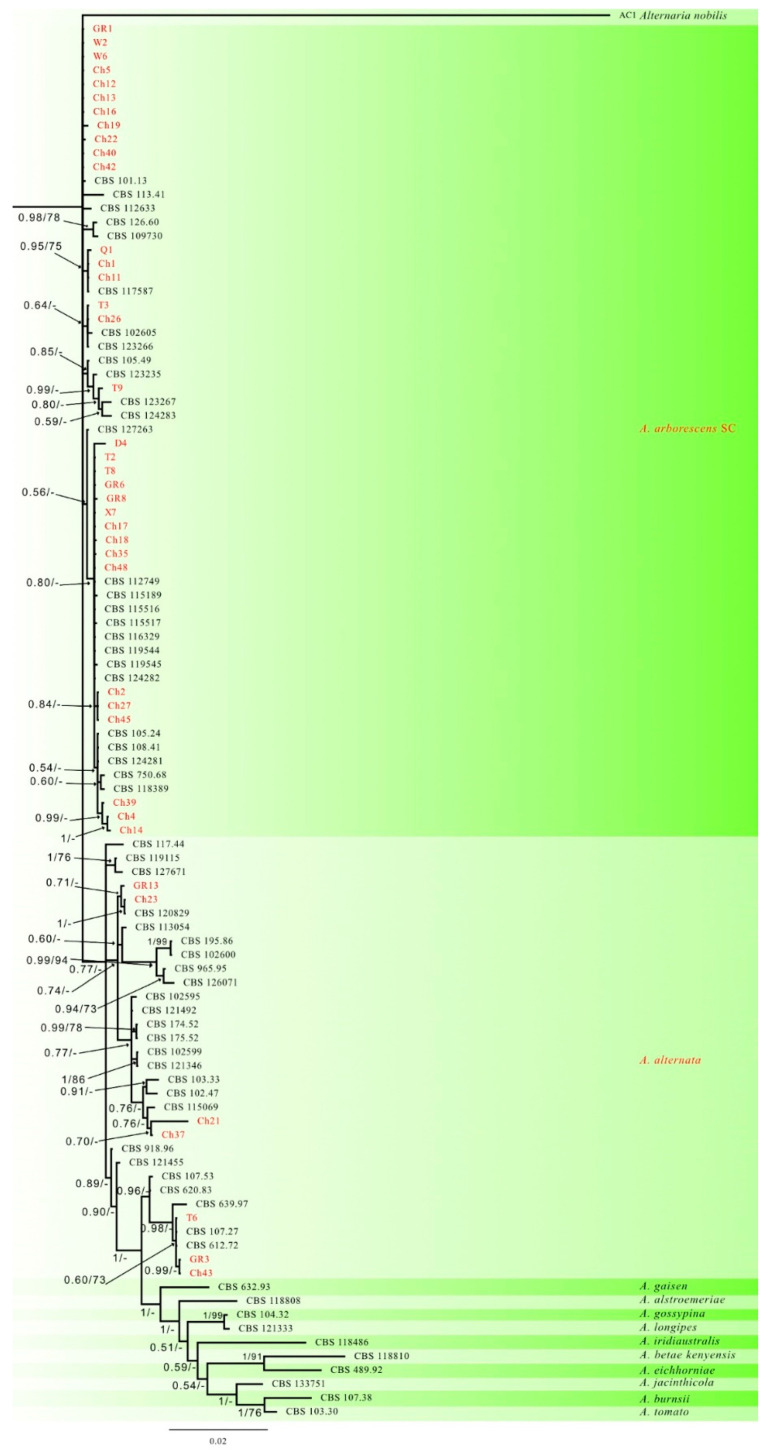
Consensus phylogram of 4947 trees resulting from a Bayesian analysis of the combined rpb2, Alta-1, endoPG and OPA 10-2 sequence alignments of the *Alternaria* species. Bootstrap support values and Bayesian posterior probability values are indicated at the nodes. The isolates obtained in this study are in red. The tree was rooted with *Alternaria nobilis* AC1.

**Figure 3 jof-09-00992-f003:**
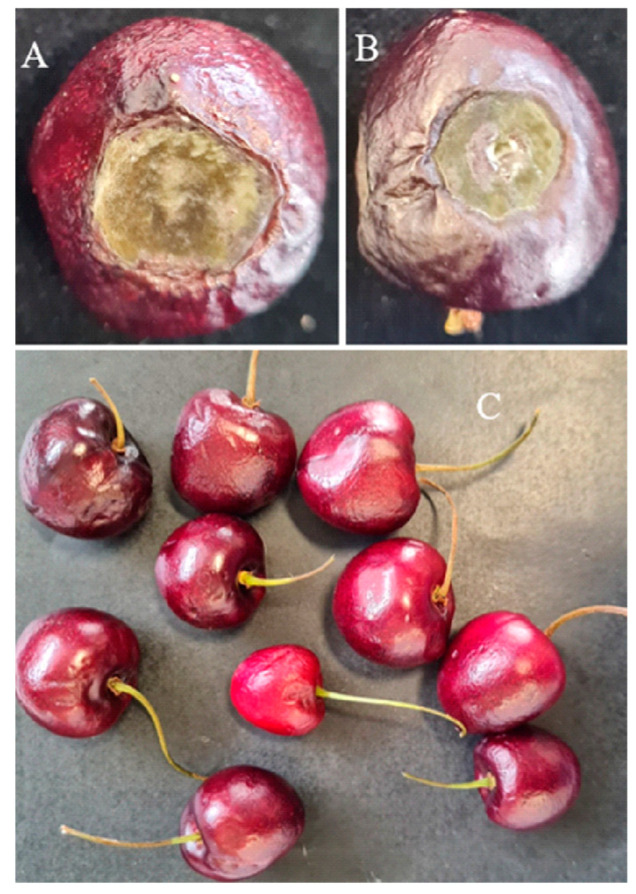
Symptoms of *Alternaria* fruit rot on sweet cherries after 10 days post inoculation (**A**) AASC (strain X7); (**B**) *A. alternata* (strain Ch23); (**C**) healthy control.

**Table 1 jof-09-00992-t001:** Strain name, species identification, year of isolation, colony growth (cm), size of conidia (length-width) and mean rot diameter obtained with the pathogenicity test for the strains isolated from sweet cherry fruits.

Isolate Name	Fungal Species	Year of Isolation	Cherry Cultivar	Colony Growth (cm ± SD)	Size of Conidia (Length × Width; μm)	Rot Diameter (mm ± SD) ^1^
D4	AASC	2020	Kordia	4.94 ± 0.59	21.73 × 11.31	12.06 ^b–d^ ± 1.52
T2	AASC	2020	Ferrovia	5.14 ± 0.28	20.13 × 11.19	9.56 ^bc^ ± 1.22
T3	AASC	2020	Ferrovia	4.98 ± 0.55	21.68 × 11.23	12.21 ^bc^ ± 1.66
T6	*A. alternata*	2020	Ferrovia	4.90 ± 0.59	20.64 × 11.18	13.81 ^b–d^ ± 1.82
T8	AASC	2020	Ferrovia	4.96 ± 0.55	20.57 × 11.10	10.65 ^b–d^ ± 1.86
T9	AASC	2020	Ferrovia	5.08 ± 0.35	21.53 × 11.27	10.75 ^bc^ ± 2.31
GR1	AASC	2020	Regina	4.90 ± 0.58	21.73 × 11.24	11.39 ^b–d^ ± 1.88
GR3	*A. alternata*	2020	Regina	4.70 ± 0.51	20.83 × 11.12	9.06 ^bc^ ± 1.81
GR6	AASC	2020	Regina	4.8 8± 0.53	20.89 × 10.99	14.94 ^b–d^ ± 2.77
GR8	AASC	2020	Regina	4.94 ± 0.59	20.94 × 11.20	11.94 ^b–d^ ± 1.68
GR13	*A. alternata*	2020	Regina	4.82 ± 0.51	20.93 × 11.21	9.06 ^bc^ ± 1.45
W2	AASC	2020	Kordia	5.02 ± 0.33	21.07 × 11.16	11.06 ^b–d^ ± 1.53
W6	AASC	2020	Kordia	5.30 ± 0.41	20.43 × 11.12	11.40 ^b–d^ ± 2.29
Q1	AASC	2020	Kordia	5.00 ± 0.78	19.76 × 11.64	14.43 ^b–d^ ± 2.64
X7	AASC	2020	Kordia	4.96 ± 0.62	20.84 × 11.31	15.25 ^cd^ ± 2.94
Ch1	AASC	2021	Regina	4.70 ± 0.70	21.25 × 11.11	9.38 ^bc^ ± 1.98
Ch2	AASC	2021	Regina	5.16 ± 0.15	20.59 × 10.86	10.94 ^bc^ ± 2.59
Ch4	AASC	2021	Regina	4.94 ± 0.55	21.30 × 11.21	8.43 ^ab^ ± 2.03
Ch5	AASC	2021	Regina	4.76 ± 0.92	21.32 × 11.15	13.93 ^b–d^ ± 1.94
Ch11	AASC	2021	Regina	5.12 ± 0.13	20.75 × 11.18	9.11 ^bc^ ± 1.61
Ch12	AASC	2021	Regina	4.96 ± 0.55	20.88 × 11.02	10.71 ^bc^ ± 3.54
Ch13	AASC	2021	Regina	4.94 ± 0.59	21.16 × 11.17	9.12 ^bc^ ± 1.66
Ch14	AASC	2021	Regina	5.26 ± 0.49	20.20 × 10.84	10.06 ^b–d^ ± 1.59
Ch16	AASC	2021	Regina	5.12 ± 0.77	21.40 × 11.31	11.75 ^b–d^ ± 1.88
Ch17	AASC	2021	Regina	5.08 ± 0.74	21.32 × 11.14	17.00 ^d^ ± 2.13
Ch18	AASC	2021	Regina	4.94 ± 0.47	21.71 × 11.23	14.25 ^b–d^ ± 1.89
Ch19	AASC	2021	Regina	4.74 ± 0.81	21.10 × 11.22	12.64 ^b–d^ ± 1.61
Ch21	*A. alternata*	2021	Regina	5.06 ± 0.34	21.02 × 10.88	11.00 ^bc^ ± 1.98
Ch22	AASC	2021	Regina	5.00 ± 0.21	20.87 × 11.07	10.06 ^bc^ ± 1.66
Ch23	*A. alternata*	2022	Regina	5.12 ± 0.53	20.90 × 11.03	11.50 ^b–d^ ± 1.90
Ch26	AASC	2022	Regina	5.18 ± 0.48	21.04 × 11.12	11.94 ^b–d^ ± 1.87
Ch27	AASC	2022	Regina	4.86 ± 0.77	21.12 ×11.09	9.25 ^bc^ ± 1.80
Ch35	AASC	2022	Regina	4.98 ± 0.41	21.03 × 11.16	10.78 ^b–d^ ± 1.93
Ch37	*A. alternata*	2022	Regina	4.52 ± 0.78	21.12 × 11.19	9.17 ^bc^ ± 1.52
Ch39	AASC	2022	Regina	4.74 ± 0.57	21.07 × 10.98	11.50 ^bc^ ± 1.93
Ch40	AASC	2022	Regina	5.18 ± 0.26	21.12 × 11.11	10.50 ^bc^ ± 1.74
Ch42	AASC	2022	Regina	5.04 ± 0.27	20.94 × 11.03	12.06 ^b–d^ ± 1.52
Ch43	*A. alternata*	2022	Regina	4.68 ± 0.68	21.05 × 11.01	10.88 ^bc^ ± 1.23
Ch45	AASC	2022	Regina	4.92 ± 0.59	21.20 × 11.29	10.81 ^bc^ ± 1.94
Ch48	AASC	2022	Regina	4.88 ± 0.50	21.11 × 11.05	9.33 ^bc^ ± 1.97
Healthy control	-	-	Regina	-	-	0 ^a^

^1^ Values are mean of the rot diameter on nine cherry fruits. Values with the same letter are not different according to Tukey’s test (*p* ≤ 0.05).

## Data Availability

Not applicable.

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
