# Peer review of "Molecular Characterization and Pathogenicity of Alternaria spp. Associated with Black Rot of Sweet Cherries in Italy"

_jof, 2023, doi:10.3390/jof9100992_

Round 1

Reviewer 1 Report

The manuscript entitled with the “Molecular Characterization and Pathogenicity of Alternaria spp. associated with Black Rot of Sweet Cherries in Italy” was identified alternaria species morphologically and molecularly. In the research 40 alternaria isolates were obtained but many of these isolates are similiar. Only two isolates were different form the others. I think that it was unneccessary to seguenced of all isolates.

In the abstract, it should be provide information about the morphological characteristics of the described Alternaria spp. Please added information about this finding

In Introduction section,

Please rewrite the last sentences as “The aims of this study were to isolate and identify the Alternaria species associated with black rot of sweet cherries based on morphological and phylogenetic analysis, and also to confirm their pathogenicity”

Please the sentence in the manuscript “Alternaria spp. are distributed allover the world and may infect over 4,000 plant species [16,18,19], such as vegetables, cereals and fruit trees in field and during storage [20–24]” the references given below  should cited

Dogan, A., Cat, A., Catal, M., & Erkan, M. (2018). First report of Alternaria alternata causing postharvest decay in fig (Ficus carica L. cv. Bursa Siyahi) fruit in Turkey. J. Biotechnol280, S32-S91.

Ustun, R., Cat, A., Uzun, B., & Catal, M. (2019). First Report of Alternaria alternata Causing Leaf Spot Disease on Soybean (Glycine max) in Antalya Province of Turkey. Plant Disease103(12), 3284.

In Materials and Methods section,

A. alternata and A. arborescens species complex (AASC) in terms of morphological definition were not stated clearly how these species were identified in the manuscript. Also the authors were not cited any reference. Please give the reference for subheadings 2.1 and 2.2.

Subheading 2.3 the PCR reactions volume was not give the correct. The sentece was rewrite  “PCR were carried out in a total volume of 25 μL, which contained: 2.5 μL of 10x buffer, 0.5 μL of MgCl2, 0.75 μL of dNTPs (10 mM), 1 μL of each primer (10 mM), 0.2 μL of Taq DNA polymerase (Invitrogen, USA)” Please give the reactions volume as mM. In addition PCR conditions should be stated in the manuscript. Please rewrite the sentence “PCR cycling conditions were performed according to [38] for rpb2 and Alt-a1, and to [28] for endoPG and OPA 10-2”

Please subheadings “2.4 Pathogenicity assay” section  add a reference.

In results section,

Please add a scale bar to the conidia (e-f) in Figure 1.

Which variety was used as a negative control. Please add Table 1.

Resolution of Figure 2 is low. Please change this and add in bootsraps value. Furthermore In the study, although it was reported that 7 isolates were identified as A. alternata, in Figure 2 shows that 6 isolates were within the A. alternata group. Please explain this situation.

The sentence “Our study demonstrates that sweet cherries are a susceptible host of AASC. Moreover, we reported for the first time the presence of AASC as agent of black rot on sweet cherry fruits in Italy, and we characterized the pathogen population”  please delete the “and we characterized the pathogen population”

Please delete this sentence “Due to the problems in identification of Alternaria spp. at species level through morphological and molecular analysis, future studies should focus on combination of pathological, morphological, and biochemical properties.”

Overall, only 40 isolates were identified and also majority of them were similar. I think that the other identified isolates should be added in the manuscript.  

Minor editing of English language required. 

Author Response

Thank you for the review of our manuscript. You have done an excellent job in reviewing this manuscript. The comments were very helpful in improving the manuscript. We have revised the manuscript according to the comments from you. The changes made in the revised manuscript are colored as red.

The manuscript entitled with the “Molecular Characterization and Pathogenicity of Alternaria spp. associated with Black Rot of Sweet Cherries in Italy” was identified alternaria species morphologically and molecularly. In the research 40 alternaria isolates were obtained but many of these isolates are similiar. Only two isolates were different form the others. I think that it was unneccessary to seguenced of all isolates.

We identified 40 Alternaria spp. (33 A. arborescens species complex and 7 A. alternata) during 3 years from black rotted cherry fruits. We tried to identify Alternaria spp. at species level and, as the morphological identification is difficult and unclear, we decided to sequence all isolates. We were able to identify A. alternata and AASC. According to previous studies, A. alternaria was the only pathogen of black rot of cherry fruits in China and now recently reported in Chile. The novelty of the manuscript is about the fact that we identified for the first time AASC on cherry. However, we will keep in mind your suggestions for the future study.

In the abstract, it should be provide information about the morphological characteristics of the described Alternaria spp. Please added information about this finding

Response: We added the morphological characteristics of the isolates of Alternaria spp. (See in revised abstract please)

In Introduction section, please rewrite the last sentences as “The aims of this study were to isolate and identify the Alternaria species associated with black rot of sweet cherries based on morphological and phylogenetic analysis, and also to confirm their pathogenicity”

Response: According to your suggestion, we have now rephrased and corrected the sentence (See in revised introduction please).

Please the sentence in the manuscript “Alternaria spp. are distributed allover the world and may infect over 4,000 plant species [16,18,19], such as vegetables, cereals and fruit trees in field and during storage [20–24]” the references given below  should cited

Dogan, A., Cat, A., Catal, M., & Erkan, M. (2018). First report of Alternaria alternata causing postharvest decay in fig (Ficus carica L. cv. Bursa Siyahi) fruit in Turkey. J. Biotechnol280, S32-S91.

Ustun, R., Cat, A., Uzun, B., & Catal, M. (2019). First Report of Alternaria alternata Causing Leaf Spot Disease on Soybean (Glycine max) in Antalya Province of Turkey. Plant Disease103(12), 3284.

Response: We have cited the references suggested by you.

In Materials and Methods section,

  1. alternataand A. arborescens species complex (AASC) in terms of morphological definition were not stated clearly how these species were identified in the manuscript. Also the authors were not cited any reference. Please give the reference for subheadings 2.1 and 2.2.

Response: We have identified the Alternaria isolates according to Simmons (2007). We included more information in the result section as well. We also gave references for subheading 2.1 and  2.2.

Subheading 2.3 the PCR reactions volume was not give the correct. The sentece was rewrite  “PCR were carried out in a total volume of 25 μL, which contained: 2.5 μL of 10x buffer, 0.5 μL of MgCl2, 0.75 μL of dNTPs (10 mM), 1 μL of each primer (10 mM), 0.2 μL of Taq DNA polymerase (Invitrogen, USA)” Please give the reactions volume as mM. In addition PCR conditions should be stated in the manuscript. Please rewrite the sentence “PCR cycling conditions were performed according to [38] for rpb2 and Alt-a1, and to [28] for endoPG and OPA 10-2”

Response: According to your suggestions, we have now rephrased and corrected the sentences and  we also gave the PCR cycling condition in manuscript (See in revised Subheading 2.3 please).

Please subheadings “2.4 Pathogenicity assay” section add a reference.

Response: We have added the reference (See in revised Subheading 2.4 please).

In results section,

Please add a scale bar to the conidia (e-f) in Figure 1.

Response: We have added the scale bar to the conidia (e-f) in Figure 1.

Which variety was used as a negative control. Please add Table 1.

Response: We used the Regina variety as a negative control. We have added the variety name in Table 1

Resolution of Figure 2 is low. Please change this and add in bootsraps value. Furthermore In the study, although it was reported that 7 isolates were identified as A. alternata, in Figure 2 shows that 6 isolates were within the A. alternata group. Please explain this situation.

Response: Thank you for your keen observation. We have changed the resolution of Figure 2. We are agree with worthy reviewer about addition of bootstrap value. However, our way of presentation is also effective and acceptable.

Yes, in our study we identified 7 A. alternata isolates. We didn’t highlight the isolate ch 21 in red (just like other isolates) by mistake now we have highlighted it in A. alternata section. (See in revised Figure 2 please).

The sentence “Our study demonstrates that sweet cherries are a susceptible host of AASC. Moreover, we reported for the first time the presence of AASC as agent of black rot on sweet cherry fruits in Italy, and we characterized the pathogen population”  please delete the “and we characterized the pathogen population”

Please delete this sentence “Due to the problems in identification of Alternaria spp. at species level through morphological and molecular analysis, future studies should focus on combination of pathological, morphological, and biochemical properties.”

Response: We have deleted the both sentences (“and we characterized the pathogen population” and “Due to the problems in identification of Alternaria spp. at species level through morphological and molecular analysis, future studies should focus on combination of pathological, morphological, and biochemical properties.”) According to your suggestions

R1: Overall, only 40 isolates were identified and also majority of them were similar. I think that the other identified isolates should be added in the manuscript.

Response: We have collected 40 isolates of Alternaria during 3 years. We were unable to add other identified isolates in the manuscript. But we will follow your useful suggestions in future studies. A sentence was added in the Discussion section.

Reviewer 2 Report

The research object is interesting, but there are several flaws.

Abstract and key words. Aim of research should be formulated more precise: “to isolate” is technical action, not part of study.

“Black rot” should be deleted from key words, because it is mentioned in the title.

Introduction. The first paragraph should be shortened, there is no need to tell about the nutritional value of cherries. Therefore, perhaps, the first three or four references are not necessary. Also, second paragraph is weakly related to object of research.

I think, Alternaria species, which have been found in cherry should be mentioned, if there are no information, it should be highlighted.

Methods and materials.

Did all isolates form conidia? It is very difficult to believe, because usually part of Alternaria isolates do not sporulate also on potato-carrot agar.

Results. Abbreviation AASC should be explained in the body text, it is not enough to explain it in abstract.

Table 1 is too long, data should be summarized. Authors have evaluated different morphological traits (in accordance with described methods), but only one sentence was written in results: “colonies were dark greyish in the center with white margins”. Did all isolates have the same morphological traits? It is difficult to believe, because usually morphological traits of Alternaria vary significantly.

Figure 3 shows fruits inoculated with both species of Alternaria – symptoms are slightly different. Is it regularity or coincidence? Why were chosen exactly these fruits?

Discussion. Discussion is very weak. Possible occurrence of other Alternaria species are not discussed. Identification of Alternaria species is complicated, but authors have not discussed possibility to use other genes. For example, others researchers have used 8 primers’ pairs.

Author Response

The research object is interesting, but there are several flaws.

Dear reviewer, thank you for the review of our manuscript. We thank you for spending your precious time on this paper. You have done an excellent job in reviewing this manuscript. The comments were very helpful in improving the manuscript. We have now revised whole manuscript according to the comments from you. Our point-by-point response to your comments are given below and included in the revised paper as inserts (marked the changes as red in manuscript).

Abstract and key words. Aim of research should be formulated more precise: “to isolate” is technical action, not part of study.

Response: Thank you. We have rephrased and corrected the sentences the aim of the research in abstract.

“Black rot” should be deleted from key words, because it is mentioned in the title.

Response: We have deleted the “Black rot” from key words

Introduction. The first paragraph should be shortened, there is no need to tell about the nutritional value of cherries. Therefore, perhaps, the first three or four references are not necessary. Also, second paragraph is weakly related to object of research.

I think, Alternaria species, which have been found in cherry should be mentioned, if there are no information, it should be highlighted.

Response: According to your suggestions we have made changes in the introduction section. Although, little information is available about black rot caused by Alternaria spp. on sweet cherry fruits , but we have added the relevant data about it in introduction. (See in revised Subheading 2.4 please).

Methods and materials.

Did all isolates form conidia? It is very difficult to believe, because usually part of Alternaria isolates do not sporulate also on potato-carrot agar.

Response: Yes, our all Alternaria isolates produced conidia with little or no variation. Our results were similar to previous studies by Prencipe et al. 2023; ÅžimÅŸek et al. 2022.

Results. Abbreviation AASC should be explained in the body text, it is not enough to explain it in abstract.

Response: We have made changes in the body text (See revised introduction please)

Table 1 is too long, data should be summarized. Authors have evaluated different morphological traits (in accordance with described methods), but only one sentence was written in results: “colonies were dark greyish in the center with white margins”. Did all isolates have the same morphological traits? It is difficult to believe, because usually morphological traits of Alternaria vary significantly.

Response: We have shortened the table, but we have included more information about colony morphology. We agree with the reviewer that morphological traits of Alternaria vary significantly, but in our study the isolates showed little variation. Our results were similar to previous studies as well. (See results, table 1 and discussion please)

Figure 3 shows fruits inoculated with both species of Alternaria – symptoms are slightly different. Is it regularity or coincidence? Why were chosen exactly these fruits?

Response: In our pathogenicity assay symptoms of A. alternata and A. arborescens species complex strains are slightly different. It is not coincidence because in our study strains of A. arborescens SC were more virulent as compared A. alternata. To fulfill the kotchs postulates we have chosen the cherry fruit (Regina)  to know that whether they produced the symptoms just like original infected cherry fruit or not.

R2: Discussion. Discussion is very weak. Possible occurrence of other Alternaria species are not discussed. Identification of Alternaria species is complicated, but authors have not discussed possibility to use other genes. For example, others researchers have used 8 primers’ pairs.

Response: Many thanks. We have now revised and improved the discussion section (See in revised discussion please).

Round 2

Reviewer 1 Report

I thank the authors for accepting the suggested changes. I wish them success in their future study

Reviewer 2 Report

I think, the manuscript is improved significantly. I still wonder about similarity of isolates and conidia in all, but these are your results.